**A new method to identify flux ropes in space plasmas**
Shiyong Huang[1], Pufan Zhao[1], Jiansen He[2], Zhigang Yuan[1], Meng Zhou[3], Huishan
Fu[4], Xiaohua Deng[3], Ye Pang[3], Dedong Wang[1], Xiongdong Yu[1], Haimeng Li[4], Roy
Torbert[5], and James Burch[6]
[1] School of Electronic Information, Wuhan University, Wuhan, China
[2] School of Earth and Space Sciences, Peking University, Beijing, China
[3] Institute of Space Science and Technology, Nanchang University, Nanchang, China
[4] School of Space and Environment, Beihang University, Beijing, China
[5] University of New Hampshire, Durham, New Hampshire, USA
[6] Southwest Research Institute, San Antonio TX, USA
**Abstract**
Flux ropes are frequently observed in the space plasmas, such as magnetosphere,
magnetosheath, and solar wind etc., and play an important role in the reconnection
process and mass and flux transportation. One usually used bipolar signature and
strong core field to identify the flux ropes. We propose here one new method to
identify flux ropes based on the correlations between the variables of the data from
in-situ spacecraft observations and the target-function-to-be-correlated (TFC) from
the ideal flux rope model. Through comparing the correlation coefficients of different
variables at different time and scales, and performing weighted average technique,
this method can derive the scales and locations of the flux ropes. We discuss the
limitation of our method and also compare it with other methods.

**1. Introduction**
Flux ropes, as one universal structure in the space plasma, are formed as a helical
magnetic structure with magnetic field lines wrapping and rotating around a central
axis (e.g., *Hughes and Sibeck*, 1987; *Slavin et al.*, 2003; *Zong et al.*, 2004; *Zhang et*
*al.*, 2010). It is generally believed that flux ropes can be generated by magnetic
reconnection in the eruptive energy processes, such as rapid variations of the



reconnection rate at a single X-line (e.g. *Nakamura and Scholer*, 2000; *Fu et al.*,
2013), multiple X-line reconnection (e.g. *Lee et al.*, 1985; *Deng et al.*, 2004). Flux
ropes play important roles in dissipating magnetic energy and controlling the
microscale dynamics of magnetic reconnection (e.g., *Drake et al.*, 2006; *Daughton et*
*al.*, 2007; *Fu et al.*, 2017). These structures are frequently observed and widely
studied recently in the magnetosphere, magnetosheath and solar wind (e.g. *Hu and*
*Sonnerup*, 2001; *Slavin et al.*, 2003; *Zong et al.*, 2004; *Zhang et al.*, 2010; *Huang et*
*al.*, 2012, 2014a, 2014b, 2015, 2016a, 2016b; *Rong et al.*, 2013). Many works have
tried to model flux rope from *in-situ* measurements based on the force-free
constant-alpha flux rope (e.g., *Lepping et al.*, 1990), non-force-free model (e.g.,
*Hidalgo et al.*, 2002), or the Grad-Shafranov equilibrium (e.g., *Hu and Sonnerup*,

42    2002).


Flux ropes embedded in current sheet are characterized by the bipolar signature of the
normal component of magnetic field, strong core field in the axis direction, and
enhancement in magnetic field strength. Therefore, one used negative-positive
(positive-negative) bipolar signature of the south-north magnetic field component in
the earthward (tailward) flow with an enhancement in the cross-tail component and
strength of magnetic field to identify flux rope in the magnetotail (e.g., *Slavin et al.*,
2003; *Huang et al.*, 2012). At the magnetopause, the bipolar variation is usually along
the sun-earth direction, and the core field is typically along the dawn-dusk direction
(e.g., *Zhang et al.*, 2010). However, flux ropes in the magnetosheath, which has been
reported recently by MMS (*Huang et al.*, 2016b), can move in any directions due to
the large fluctuations of the shocked solar wind. This leads to difficulty in identifying
the flux ropes there.

Several attempts are tried to survey flux ropes in the Earth's magnetotail by eyes
based on their signatures, such as bipolar variation of north-south magnetic field (e.g.,
*Richardson et al.*, 1987; *Slavin et al.*, 2003). Also, some methods are proposed to
automatically in some degrees survey the flux rope or flux transfer events (FTEs) via



bipolar field deflections (e.g., *Kawano and Russell*, 1996; *Vogt et al.*, 2010; *Jackman*
*et al.*, 2014; *Smith et al.*, 2016). *Karimabadi et al.* (2009) have applied data mining
technique (MineTool) to search FTEs using magnetic field and plasma data. Recently,
*Smith et al.* (2017) developed a method to automatically detect cylindrically
symmetric force-free flux ropes in the magnetotail only using magnetic field data.
That method first locates the significant deflections in the north-south magnetic field
component with peaks in the dawn-dusk component or total field. Then, the
candidates are using Minimum Variance Analysis (MVA) to determine a local
coordinate system. Finally, the candidates are fitted by a fore-free model to determine
whether they belong to flux ropes or not.

For some flux ropes with short duration, the plasma data have not enough high time
resolution or even worse are not available. Thus the identification of flux ropes relies
heavily on the magnetic field data. All aforementioned automatical methods are a bit
complex, or require plasma data. Therefore, to identify flux rope only using the
magnetic field data from single spacecraft, we propose a new and simple method
based on the correlation coefficients between the signal and the ideal model of flux
rope to identify flux ropes in space plasmas. The paper will be presented as follows:
an introduction of the method in section 2; the test of the method on artificial data
from the model in section 3; the applications of the method on the Cluster and MMS
data in section 4; summary is given in section 5.

**2. Approach**
In this section, we simply introduce our method.
Firstly, we derive target-function-to-be-correlated (TFC) from the ideal model of flux
rope. Considering the variable and complicated observed flux ropes, we use the ideal
non-force-free model of flux rope proposed by *Elphic and Russell* (1983), named as
Elphic and Russell (E-R) Model because most of flux ropes with nonnegligible
perpendicular current are not consistent with force-free model (e.g., *Hidalgo et al.*,
2002; *Zong et al.*, 2004; *Zhang et al.*, 2010; *Borg et al.*, 2012; *Huang et al.*, 2012,



2016b). This model is constructed with an intense core field inside of flux rope, which
is shown in Figure 1. The equation of this model in the cylindrical coordinate (Y is
defined as the axis orientation of flux rope) can be modified as below:
$$\begin{cases} B_y = B(r)\cos(\alpha(r)) \\ B_\varphi = B(r)\sin(\alpha(r)) \\ B(r) = B_0\exp(-r^2/b^2) \end{cases} \tag{1}$$

Where $\alpha(r) = \pi/2(1-\exp(-r^2/a^2))$, $B_y$ is the core field component, $B_0$, $a$, and $b$ are the
constant, $r$ is the radial distance to the flux rope center.

Figure 1 shows sketched diagram of the cylindrical flux rope from E-R model. For
convenience, the rectangular coordinate is used in our analyses (shown in Figure 1). Y
is the axis orientation of the flux rope, and the X-Z plane is the cross-section
perpendicular to the axis orientation. X can be treated as sun-earth orientation, Y is
the dawn-dusk orientation, and Z is similar to the south-north orientation in the
magnetotail. If the one spacecraft cross the flux rope following the red path in Figure
1, $B_z$ component will be characterized as bipolar signature, and $B_y$ component has
strong peak.

Figure 2 shows the observations when one virtual spacecraft cross the ideal flux rope
(see spacecraft path in Figure 1). Here we assume the scale of flux rope as one unit,
and 1 unite/s of moving speed of the spacecraft, thus set $a = 0.735$ s and $b = 0.735$ s,
$B_0 =10$ nT, and use the $B_z$ as the bipolar variation component, $B_y$ as the core field
component, $B_t$ as the total magnetic field. The center of the flux rope is located at 2.5
s. One can see the $B_z$ bipolar signature, and the peak of core field and total magnetic
field inside the flux rope.

Considering the previous observations, in which the $B_z$ component during the crossing
of the flux rope usually does not reach zero like that shown in Figure 2a, we cut out
one part of the ideal flux rope as the TFC which is shown in Figure 3. The TFC is
similar to the sinusoidal function when one performs Fast Fourier Transform (FFT)





analysis. We only used two components ($B_y$ and $B_z$) and magnetic strength ($B_t$) as the
TFC since only $B_z$ and $B_y$ components and $B_t$ have very obvious typical feature
usually from in-situ measurements (i.e., $B_z$ has bipolar signature, $B_y$ is strong core
field, and $B_t$ has peak inside flux rope), and $B_x$ component has not common feature
from observation viewpoint (e.g., *Slavin et al.*, 2003; *Huang et al.*, 2014a).

Secondly, we calculate the correlation coefficients between the signal and the TFC at
different time and different scales. Before calculating the correlation coefficients, the
amplitude of the TFC will be estimated from the signal. For example, the maximum
value of $B_t$ during the time interval is used as the amplitude of $B_t$ in the TFC. The
sliding time window is used in the calculation of the correlation coefficients. The
calculated results of correlation coefficients are similar to the power spectral densities
by FFT that displays the power spectral density at different time and different
frequency. The higher values of the correlation coefficients, the more suitable for the
description of the model on the signal.

Thirdly, we compare the correlation coefficients of the bipolar variation component
$B_z$, core field component $B_y$, and total magnetic field $B_t$, and find out the high
correlations (larger than the given threshold) at the same time and the same scale.
This is due to that the bipolar signature in $B_z$, the enhancements of core field $B_y$ and
magnetic strength $B_t$ should appear simultaneously with the same duration when one
spacecraft cross the flux ropes.

Fourthly, we infer the location and the scale of the flux ropes based on the weighed
average (it will be shown later), and the amplitude from minimum to maximum values
of the bipolar variation.

**3. Model test**
One test is performed on the artificial data from E-R model plus the random noise.
Figure 4 presents the test results. The test artificial data is shown in Figure 4a where



the noise is 10% of the amplitude of the flux rope. A series of the calculations are
carried on $B_z$, $B_y$ and $B_t$ to obtain the correlation coefficients. One should point out
that the absolute values of the correlation coefficients of $B_z$ and $B_y$ are given in Figure
4b and 4c respectively, because the bipolar structure can be positive-negative or
negative-positive variation and the core field can be positive or negative. It can be
seen that the correlation coefficients are largest at the scale of 0.6 ~ 1.5 s during the
crossing of the flux rope (around *time* ~ 3.5 s).

We set the threshold as 0.9 to represent the results in Figure 5 where only the
correlation coefficients with > 0.9 are displayed with black shadows. All correlation
coefficients of the three variables have peaks at the *time* ~ 3.5 s with the time scale ~
1 s. We use the weighted average technique (shown below) to identify the flux rope
and estimate its time scale.
$\quad \tau = \sum coef_i \times \tau_i / \sum coef_i$ (2)
where $coef_i$ is the correlation coefficient at time scale $\tau_i$.

Figure 5e shows the estimated results. The crossing of the flux rope is marked with "1"
and the duration is its scale, the center of the flux rope is at the center of the line. In
this test, the scale is estimated as 1.039 s, the location is 3.496 s. The amplitude is
estimate as 4.43 nT from minimum to maximum values of the bipolar variation.
Aforementioned sets, one can estimate the error of the scale as 3.9%, i.e.,
(1.039-1.0)/1.0 = 3.9%. Therefore, our method can successfully identify the flux rope,
and estimate its scale, location and amplitude.

**4. Application**
In this section, we apply our new method to the spacecraft measurements in the
magnetosheath and the magnetotail.

4.1 Flux rope in the magnetosheath
Flux ropes are successfully identified in the magnetosheath using the unprecedented



high resolution data from Magnetospheric Multiscale (MMS) (*Burch et al.*, 2016)
mission (*Huang et al.*, 2016b). Their observations demonstrate that highly dynamical,
strong wave activities and electron-scale physics occur in the magnetosheath ion-scale
flux ropes. Figure 6 gives the observations of ~14 s from MMS2 on 25 Oct 2015 and
the test results of our method. Similar to the model test, we use the same variables to
present the components of the bipolar variation, core field and total magnetic field
after transformed to minimum variable analysis (MVA) analysis (*Huang et al.*,
2016b). The threshold of the correlation coefficients is also set as 0.9 in Figure 6. We
can see that the correlation coefficients of the three variables (Figure 6b-6d) only have
high values at the same time around *time* = 5.5 s, implying that one flux rope is
identified by this method. Based on the weighted average method in equation (2), the
time scale of the flux rope is 1.11 s, and its central location is at 5.38 s. The amplitude
is estimate as 115 nT. All these results are consistent with previous findings from
multi-spacecraft data in *Huang et al.* (2016b).

4.2 Flux rope in the magnetotail
Flux ropes are frequently observed in the magnetotail, and play an important role
during magnetic reconnection and magnetotail dynamic (e.g., *Slavin et al.*, 2003;
*Zong et al.*, 2004; *Chen et al.*, 2008; *Huang et al.*, 2012, 2016a; *Fu et al.*, 2015, 2016).
Chen et al. (2008) have identified several flux ropes filled with energetic electrons
during magnetic reconnection on 10 Jan 2001 by using the Cluster data. Figure 7
shows the magnetic field in GSM coordinates from the Cluster mission (*Escoubet et*
*al.*, 1997) in the magnetotail and the application results of our method. There are
several bipolar variations in $B_z$ during this time interval (Figure 7a). Figures 7b-7d
present the correlation coefficients (larger than 0.9 of the threshold) of the three
variables. Here we try to identify small-scale flux ropes, so that we perform the
method only at short time scale. There are full of high correlation coefficients (grey
shadows) in Figures 6b-6d. After compare with the correlation coefficients at the
same time and same scale, our method resolves three possible flux ropes in Figure 7e.
The results are summarized in Table 1. The three structures are close to ideal flux



rope with bipolar signature in $B_z$, and peaks in core field $B_y$ and total magnetic field $B_t$.
All three flux ropes identified by our method have been reported in *Chen et al.*

211    (2007).


We should point out that our method only can identifies the flux rope and derives its
duration. If the plasma velocity data is available, then we can estimate the actual
spatial scale of the flux rope. If multi-spacecraft data are available for the time
interval of interest, one can derive the size, the orientation, and the motion of the flux
rope using by the multi-spacecraft method such as *Sonnerup et al.* (2004), *Shi et al.*
(2005, 2006) and *Zhou et al.* (2006a, 2006b). However, the separation of the Cluster
was much lager than the size of the flux ropes on 01 October 2001, implying that one
cannot use multi-spacecraft method here.

**5. Summary and Discussion**
In summary, we developed a new method to identify flux ropes in the space plasmas.
This method is based on the correlation coefficients between the signal and the TFC
from E-R model. If the correlation coefficients of three variables ($B_z$, $B_y$ and $B_t$) of the
signal have high values of correlation coefficients at the same time and same scale,
one can deduce the existence of one flux rope and estimate its location and its time
scale (i.e., the duration). The tests on the artificial data and the in-situ realistic
spacecraft data show that our method can successfully search out the flux ropes and
obtain their locations and time scales.

Bipolar variation in $B_z$ component and the enhancement in core field and magnetic
field strength are the typical signatures for most of flux rope. But it doesn't mean that
all observations from any crossing of the spacecraft would have those signatures,
which depends on the spacecraft trajectory. However, one only can select or identify
the flux rope showing the typical signatures, and miss other flux rope not having the
typical signatures. Some special field structures may induce the similar signatures
along some special trajectories. But we thought this opportunity is too few in the





magnetotail. However, one can use the plasma measurements to rule out this
possibility.

Aforementioned attempts are used to identify flux ropes in the Earth's magnetotail by
eyes or half-automatically based on the bipolar variation of (e.g., *Richardson et al.*,
1987; *Slavin et al.*, 2003; *Kawano and Russell*, 1996; *Vogt et al.*, 2010; *Jackman et al.*,
2014; *Smith et al.*, 2016). *Karimabadi et al.* (2009) used data mining technique
(MineTool) to search flux ropes using both magnetic field and plasma data. That
method is too complex to apply in the data analysis. *Smith et al.* (2017) proposed one
method to automatically detect force-free flux ropes based on magnetic field data
from single spacecraft. In present study, we used the TFC derived from non-force-free
flux rope model to calculate the correlation coefficients with the signal, and then
compare the large correlation coefficients of different variables to identify the flux
rope. Our method is flexible and easy to apply in the in-situ spacecraft data compared
with other methods. We will quantitatively model the flux ropes identified by our
method and derive more information of the flux ropes.

We should point out that there are several limitations in our method.
1. Our method can only detect the nearly ideal cylindrical flux rope since we used
non-force-free E-R model to describe the TFC, which limits the application of this
method. The non-force-free model proposed by E-R is just one possible solution of all
the flux rope that satisfies $J \times B \neq 0$. Actually one can use the other flux rope models to
replace E-R model, and extend our method to identify the flux ropes.

2. If the flux ropes are not well regular, there are large time deviations between $B_z$, $B_y$
and $B_t$ which will lead to miss of some flux ropes when we apply the method.

3. The threshold value of correlation coefficients can affect the results. When the
threshold value is too small that the method finds out some possible structures which
do not belong to flux ropes, or too large that the method will miss some flux ropes.






4. The correlation coefficients at small scale (especially in $B_y$ and $B_t$) could be very
large, which may affect our results. The method may find some possible structures
related to such fluctuations. We will improve this method and apply it to detect the
flux ropes in the turbulent magnetosheath in the future.


**Acknowledgement**
We thank the entire Cluster and MMS team and instrument leads for data access and
support. This work was supported by the National Natural Science Foundation of
China (41374168, 41404132, 41574168, 41674161), Program for New Century
Excellent Talents in University (NCET-13-0446), and China Postdoctoral Science
Foundation Funded Project (2015T80830). MMS Data is publicly available from the
MMS Science Data Center at http://lasp.colorado.edu/mms/sdc/. Cluster Data is
publicly    available    from    the    Cluster    Science    Archive    at
http://www.cosmos.esa.int/web/csa.

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

Khrabrov, A. V., and B. U. O. Sonnerup, DeHoffmann-Teller Analysis, in Analysis
Methods for Multi-Spacecraft Data, edited by G. Paschmann and P. W. Daly, chap. 9,
pp.221-248, Int. Space Sci. Inst., Bern, Switzerland, and Eur. Space Agency, Paris,
France, 1998.
Lee, L. C., Z. F. Fu, and S.-I. Akasofu (1985), A simulation study of forced
reconnection processes and magnetospheric storms and substorms, J. Geophys. Res.,
90(A11), 10,896–10,910.
Lepping, R. P., J. A. Jones, and L. F. Burlaga, Magnetic field structure of
interplanetary magnetic clouds at 1 AU, J. Geophys. Res., 95, 11,957-11,965, 1990.
Nakamura, M. and M. Scholer (2000), Structure of the magnetopause reconnection
layer and of flux transfer events: Ion kinetic effects, J. Geophys. Res., 105(A10),
23,179–23,191, doi:10.1029/2000JA900101.
Rong, Z. J., W. X. Wan, C. Shen, T. L. Zhang, A. T. Y. Lui, Y. Wang, M. W. Dunlop,



Y. C. Zhang, and Q.-G. Zong (2013), Method for inferring the axis orientation of
cylindrical magnetic flux rope based on single-point measurement, J. Geophys. Res.
Space Physics, 118, 271–283, doi:10.1029/2012JA018079.
Slavin, J. A., Lepping, R. P., Gjerloev, J., Fairfield, D. H., Hesse, M., Owen, C. J.,
Moldwin, M. B., Nagai, T., Ieda, A., and Mukai, T. (2003), Geotail observations of
magnetic flux ropes in the plasma sheet, J. Geophys. Res., 108, 1015–1032.
Shi, Q. Q., C. Shen, Z. Y. Pu, M. W. Dunlop, Q.-G. Zong, H. Zhang, C. J. Xiao, Z. X.
Liu, and A. Balogh, Dimensional analysis of observed structures using multipoint
magnetic field measurements: Application to Cluster, Geophys. Res. Lett., 32,
L12105, doi:10.1029/2005GL022454, 2005.
Shi, Q. Q., C. Shen, M. W. Dunlop, Z. Y. Pu, Q.-G. Zong, Z.-X. Liu, E. A. Lucek,
and A. Balogh, Motion of observed structures calculated from multi-point magnetic
field measurements: Application to Cluster, Geophys. Res. Lett., 33, L08109,
doi:10.1029/2005GL025073, 2006.
Smith, A. W., C. M. Jackman, and M. F. Thomsen (2016), Magnetic reconnection in
Saturn's magnetotail: A comprehensive magnetic field survey, J. Geophys. Res. Space
Physics, 121, 2984–3005, doi:10.1002/2015JA022005.
Smith, A. W., J. A. Slavin, C. M. Jackman, R. C. Fear, G.-K. Poh, G. A. DiBraccio, J.
M. Jasinski, and L. Trenchi (2017), Automated force free flux rope identification, J.
Geophys. Res. Space Physics, 122, 780–791, doi:10.1002/2016JA022994.
Sonnerup, B. U. O., H. Hasegawa, and G. Paschmann, Anatomy of a flux transfer
event seen by Cluster, Geophys. Res. Lett., 31, L11803, doi:10.1029/2004GL020134,

381    2004.

Zhou, X.-Z., Q.-G. Zong, Z. Y. Pu, T. A. Fritz, M. W. Dunlop, Q. Q. Shi, J. Wang,
and Y. Wei, Multiple triangulation analysis: another approach to determine the
orientation of magnetic flux ropes, Ann. Geophys., 24, 1759-1765, 2006a.
Zhou, X.-Z., Q.-G. Zong, J. Wang, Z. Y. Pu, X. G. Zhang, Q. Q. Shi, and J. B. Cao,
Multiple triangulation analysis: application to determine the velocity of 2-D structures,
Ann. Geophys., 24, 3173-3177, 2006b.
Zong, Q.-G., et al. (2004), Cluster observations of earthward flowing magnetic island



in the tail, Geophys. Res. Lett., 31, L18803, doi:10.1029/2004GL020692.
Zhang, H., et al. (2010), Evidence that crater flux transfer events are initial stages of
typical flux transfer events, J. Geophys. Res., 115, A08229, doi:10.1029/2009JA01

392    5013.







Table 1. The location, sale and amplitude of the flux ropes identified by the method.
The amplitude is defined as the values of the bipolar variation from minimum to
maximum.

| # of flux rope | 1 | 2 | 3 |
|---|---|---|---|
| Location [s] | 37.91 | 113.79 | 127.93 |
| Scale [s] | 1.99 | 2.84 | 2.05 |
| Amplitude [nT] | 9.96 | 20.49 | 12.59 |


















**Figure captions**

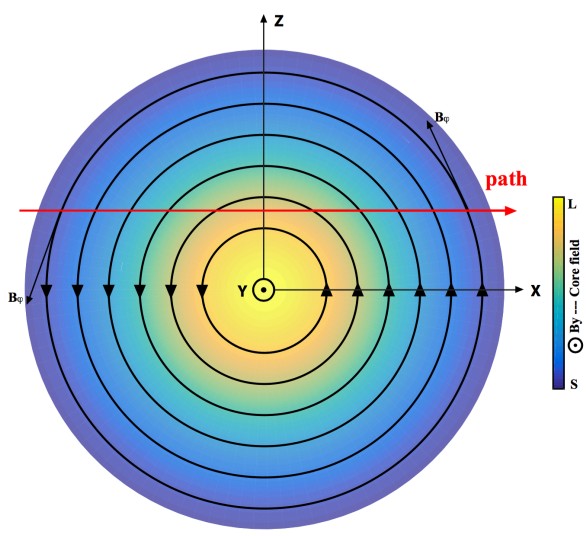


Figure 1. Sketched diagram of the cylindrical flux rope. The flux rope is right-hand
handedness structure. The black circled lines are the magnetic field lines. The red
arrow is the projection of spacecraft path. The rectangular coordinate is used in our
analyses. Y is the axis orientation of the flux rope, and the X-Z plane is the
cross-section perpendicular to the axis orientation. The core field is out-of-plane, and
the color represents the relative strength of core field (yellow: large, blue: small).



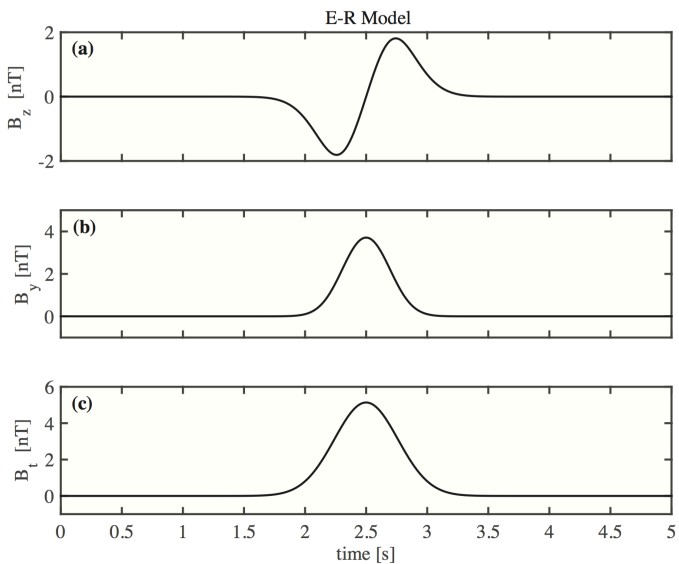


Figure 2. The three variables $B_z$ (a), $B_y$ (b), and $B_t$ (c) of the ideal cylindrical flux rope
described by E-R model.





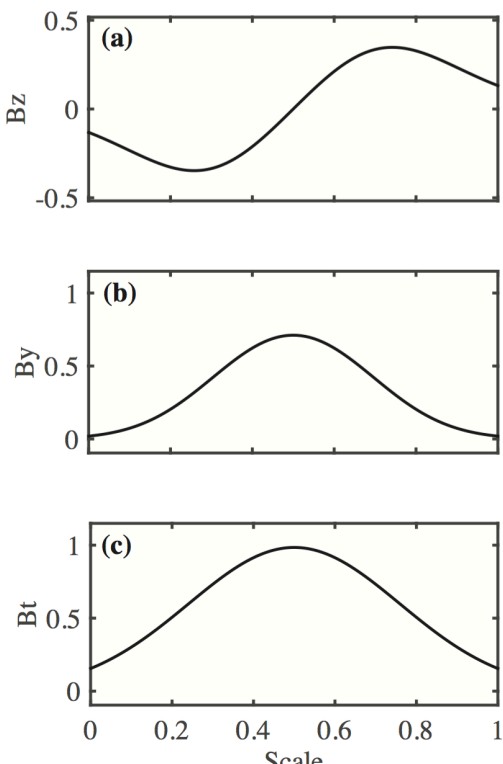


Figure 3. The target-function-to-be-correlated (TFC) derived from E-R model. The
amplitudes and scale are dimensionless.







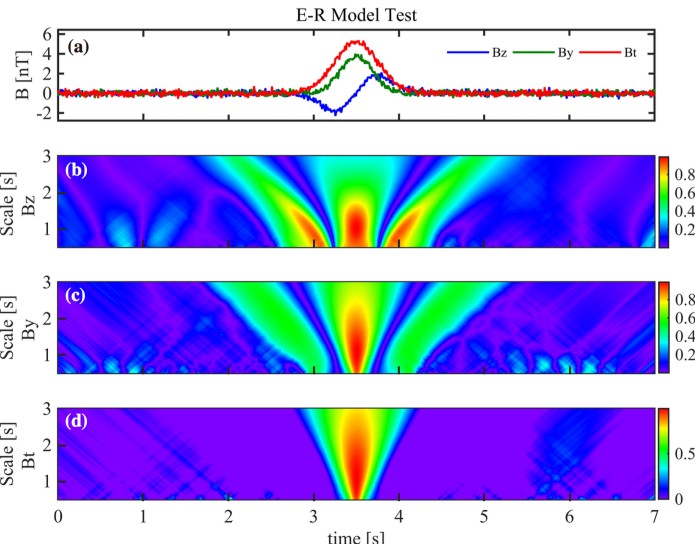


Figure 4.The test results on E-R model. (a) three variables $B_z$, $B_y$, and $B_t$ from E-R
model with 10% random noise; (b-d) the correlation coefficients between the
variables of $B_z$, $B_y$, and $B_t$ and the TFC shown in Figure 3, respectively.





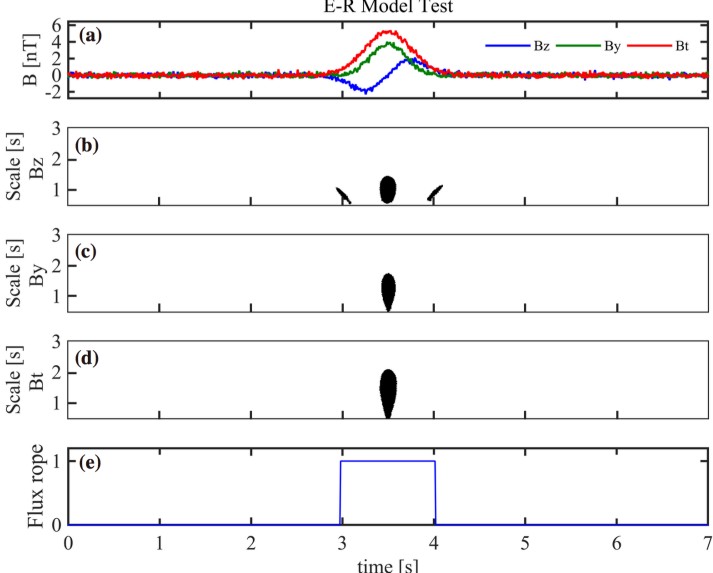


Figure 5. The test results on E-R model with a threshold 0.9. (a) three variables $B_z$, $B_y$,
and $B_t$ from E-R model with 10% random noise; (b-d) the correlation coefficients
($\geq 0.9$) between the variables of $B_z$, $B_y$, and $B_t$ and the TFC, respectively; (e) the index
when the virtual spacecraft cross the flux rope (if the spacecraft cross the flux rope,
the index is 1; if not, the index is 0). The duration of the index presents the time scale
of the flux rope.





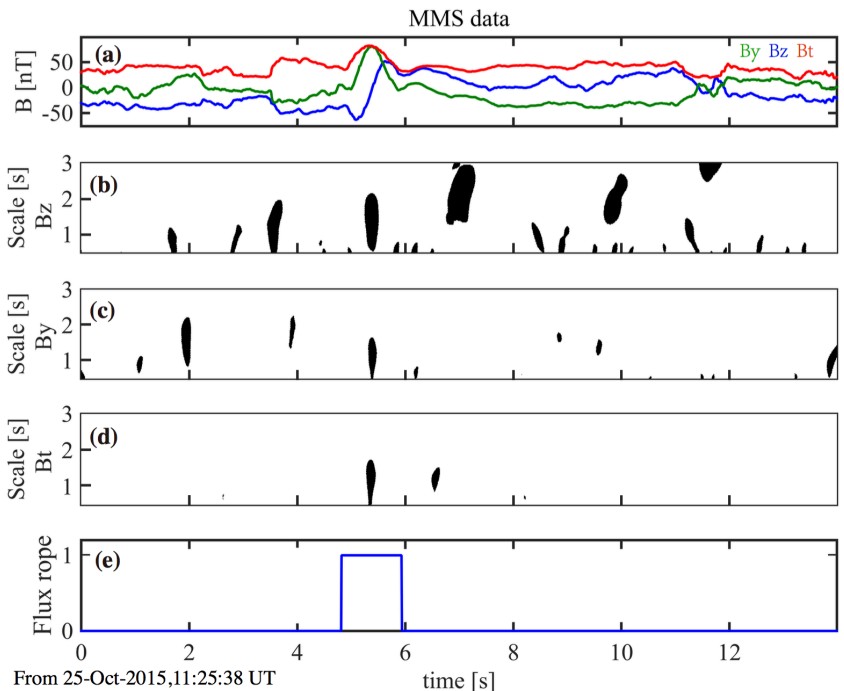


Figure 6. Testing the method on MMS data in the magnetosheath. The same format as
in Figure 5.





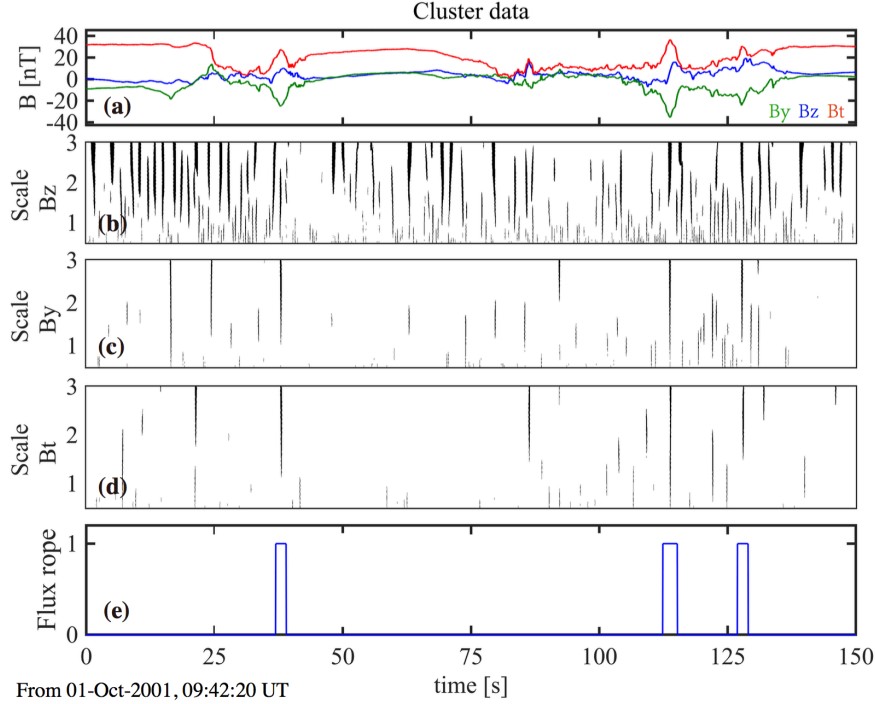

From 01-Oct-2001, 09:42:20 UT
Figure 7. Testing the method on Cluster data in the magnetotail. The same format as
in Figure 5.