# Peer review of "A new method to identify flux ropes in space plasmas"

_Annales Geophysicae, 2018_

## Referee Comment (RC1) · Anonymous Referee #1 · 22 May 2018

General Comments

This paper introduces a new method that can be used to search for flux ropes from in-situ observations. This method fits observation data into a flux rope model and uses the correlation coefficient between data and model to tell where the flux rope is. This method is important for making future studies on flux ropes easier. However, there is an important issue that needs to be clarified about this method (see the specific comments below).

Specific Comments

Line 182 and all the following parts: When the authors compare the model with real data, they did not mention how they determined the unit length for the model and the B0

to put in the model. These values must be related the data and are crucial for applying the model to the data. Therefore, the authors should explain in detail how these values are determined.

Technical corrections

Line 109: unite/s -> unit/s

Line 109: "thus set a=0.735 s ..." Here the authors use the 's' for the unit of a and b, which are length quantities. In other parts of the text the authors use 's' to mean 'seconds', a unit for time. In this line, 's' actually means 'units'. Please use a different letter for this unit.

Line 191: estimate -> estimated

Line 196: dynamic -> dynamics

Line 206: 6b-6d -> 7b-7d

Figure 4: please explain in the caption that the time scale on the vertical axes is the $\tau$ mentioned in the text.

---

## Short Comment (SC1) · 30 May 2018

1. The specific method seems simple but rather non-convincing since it will probably work for one specific type of flux rope model which is used to determine the Test Function to be Correlated (TFC). There are studies in the literature, where flux rope signatures were identified in CME structures with a lot more input models.

2. The specific method does not consider plasma data for the flux rope identification. When analysing Cluster observations, due to the highly inclined orbit of the spacecraft and the flapping motion of the magnetotail, it is possible to get magnetic field signatures that are similar to those predicted by the model but in reality are not related to a flux rope.

[Figure]

3. The paper does not mention which correlation coefficient is considered. If the Pearson correlation coefficient is calculated, then the study detects the times when there is a linear relation between the data and the TFC such as: data=A*TFC+B (where A and B constants). It is not clear in the paper that this relation is expected. If the purpose is to identify E-R flux ropes features in the data, then why not fitting the specific functions instead of correlating them? It is important to note that maximizing the correlation coefficient is not a fitting method but investigation of the linear relation between the data and the TFC.

4. How does the path of the spacecraft affect the results? Is the TFC constructed considering only one path of the spacecraft through the flux rope? Then, it is not clear how the results will not be biased due to that. For example, if different path is considered for the same flux rope, what will the method derive and how the correlation coefficients will be affected.

5. As shown in the paper (Figure 3), the TFC is a "part" of the ideal model shown in Figure 2. More specifically, TFC does not include the "edges" where the B field drops to zero. Under which criteria the specific part of the ideal model is selected. How this selection affects the determined scales?

6. The method correlates the signatures of two components and the total magnitude. Does it matter which two of the three components you use?

7. As mentioned in the paper, the correlation coefficient could be very large at small scales. Does the study consider the statistical significance of those coefficients? It is also mentioned that the method will be improved in order to be applied to data-sets in the turbulent magnetosheath. Is there a specific plan for the future?

Minor point: It probably worth adding references to previous studies that have used maximizing the correlation coefficient methods.

The above comments came from a broader discussion within the UCL/MSSL plasma

group.
* * *
Interactive
comment

---

## Short Comment (SC2) · 6 Jun 2018

The paper introduces a new technique that could be used to locate magnetic flux ropes within spacecraft magnetometer data. The correlations between signatures in two field components and the total field are used to locate the structures in data. The technique is tested using a model before being applied to example spacecraft data. The method is interesting and potentially very useful, though some of the ideas could be further developed.

Specific Comments:

1) The test with the model and additional random noise (Section 3) could be further used to benchmark the technique. Currently, the level of noise applied is very low

and (to the eye) doesn't change the signature significantly. It would be a good test to increase the value of this noise incrementally (e.g. 20

2) The dependence on the spacecraft trajectory is discussed (Line 232+), but only qualitatively. Simple tests could be performed with magnetic field models to investigate the efficacy of the method with various trajectories. This would significantly help the discussion and justification of the technique.

3) In general, the work would benefit from additional justification regarding the use of the technique. For example, in what specific ways is the method an improvement over previous attempts/survey methods (e.g. by eye searches)? What is the problem/science question that the use of this technique would help to solve? This discussion is hinted at around Line 252, but could do with development and would improve the impact of the work.

---

## Referee Comment (RC2) · Anonymous Referee #2 · 12 Jun 2018

Comments on the manuscript entitled "A new method to identify flux ropes in space plasmas" by Huang et al. Magnetic flux ropes have been regarded as an most important byproduct of magnetic field reconnection. They can raise the reconnection rate, generate energetic electrons, and change the magnetic topology. The typical signatures of a magnetic flux rope are a bipolar variation of magnetic field in the normal direction of the current sheet, and a significant enhancement of the core field as well as the total magnetic field strength. Generally, the flux ropes in the magnetopause and the magnetotail can be easily recognized from the spacecraft measurement by eyes. However, it becomes complicated while the flux ropes were located in the magnetosheath where the current distribution was turbulent. In this paper, the authors proposed a new method to identify the flux ropes in both the large-scale current sheet

and the small-scale current sheet. Furthermore, this method was tested by the Cluster and MMS observations in the magnetosheath and magnetotail. The results indicates that the method can select the flux rope effectually. In my view, the result is new and interesting, and suitable to publish in Ann. Geophys. after the following issues are considered. Line 26 flux ropes → magnetic flux ropes Line 31 Wang et al., PRL 2010 reported the first evidence of magnetic flux rope or island generated during a single X-line, should be cited here. Line 34, Wang et al., Nat. Phys. 2016 shows a clear picture of the dissipation role that the magnetic flux ropes play on. Line 103 cross → crosses

---

## Author Comment (AC1) · 31 Jul 2018

We greatly thank the reviewer for the valuable comments and suggestions that we tried to consider in the re-submission. We have revised and improved the manuscript in response to the reviewer's comments. All revised parts are marked in red in the text. Detailed answers to the comments are listed below.

Q1: General Comments This paper introduces a new method that can be used to search for flux ropes from in-situ observations. This method fits observation data into a flux rope model and uses the correlation coefficient between data and model to tell where the flux rope is. This method is important for making future studies on flux ropes easier. However, there is an important issue that needs to be clarified about

[Figure]

this method (see the specific comments below). Specific Comments Line 182 and all the following parts: When the authors compare the model with real data, they did not mention how they determined the unit length for the model and the B0 to put in the model. These values must be related the data and are crucial for applying the model to the data. Therefore, the authors should explain in detail how these values are determined.

Aw: Thanks for the referee's reminding. We used the same unit of the real data as the unit length for the model, i.e. second ('s') in our test. We revised the caption of figures accordingly.

The amplitude (B0) in the TFC is determined by the maximum value of Bt during the interval when calculate correlation coefficients.

We included this in the Line 184-187.

Q2: Technical corrections Line 109: unite/s -> unit/s Line 109: "thus set a=0.735 s . . ." Here the authors use the 's' for the unit of a and b, which are length quantities. In other parts of the text the authors use 's' to mean 'seconds', a unit for time. In this line, 's' actually means 'units'. Please use a different letter for this unit. Line 191: estimate -> estimated Line 196: dynamic -> dynamics Line 206: 6b-6d -> 7b-7d Figure 4: please explain in the caption that the time scale on the vertical axes is the $\tau$ mentioned in the text.

AW: According to the referee's suggestions, we have revised the related parts in the new version of the manuscript.

Please also note the supplement to this comment:
https://www.ann-geophys-discuss.net/angeo-2018-42/angeo-2018-42-AC1-supplement.pdf

---

## Author Comment (AC2) · 31 Jul 2018

We greatly thank the reviewer for the valuable comments and suggestions that we tried to consider in the re-submission. All revised parts are marked in red in the text.

Q: Comments on the manuscript entitled "A new method to identify flux ropes in space plasmas" by Huang et al. Magnetic flux ropes have been regarded as an most important byproduct of magnetic field reconnection. They can raise the reconnection rate, generate energetic electrons, and change the magnetic topology. The typical signatures of a magnetic flux rope are a bipolar variation of magnetic field in the normal direction of the current sheet, and a significant enhancement of the core field as well as the total magnetic field strength. Generally, the flux ropes in the magnetopause

and the magnetotail can be easily recognized from the spacecraft measurement by eyes. However, it becomes complicated while the flux ropes were located in the magnetosheath where the current distribution was turbulent. In this paper, the authors proposed a new method to identify the flux ropes in both the large-scale current sheet and the small-scale current sheet. Furthermore, this method was tested by the Cluster and MMS observations in the magnetosheath and magnetotail. The results indicates that the method can select the flux rope effectually. In my view, the result is new and interesting, and suitable to publish in Ann. Geophys. After the following issues are considered. Line 26 flux ropes → magnetic flux ropes Line 31 Wang et al., PRL 2010 reported the first evidence of magnetic flux rope or island generated during a single X-line, should be cited here. Line 34, Wang et al., Nat. Phys. 2016 shows a clear picture of the dissipation role that the magnetic flux ropes play on. Line 103 cross → crosses

AW: We have revised manuscript according to the referee's suggestions.

Please also note the supplement to this comment:
https://www.ann-geophys-discuss.net/angeo-2018-42/angeo-2018-42-AC2-supplement.pdf

---

## Author Comment (AC3) · 31 Jul 2018

We greatly thank the UCL/MSSL plasma group for the valuable comments and suggestions that we tried to consider in the re-submission. We have revised and improved the manuscript in response to the comments. All revised parts are marked in red in the text. Detailed answers to the comments are listed below.
AW:Thanks for your comments. In our method, we utilized the non-force-free model based the fact that most of flux ropes are not consistent to the fore-free model (may be close to quasi force-free). Our method can be also applied easily to other flux rope models.

2. The specific method does not consider plasma data for the flux rope identification. When analysing Cluster observations, due to the highly inclined orbit of the spacecraft and the flapping motion of the magnetotail, it is possible to get magnetic field signatures that are similar to those predicted by the model but in reality are not related to a flux rope.

AW: We proposed the method to identify flux rope only using the magnetic field data from single spacecraft in our paper because the plasma measurements are not available sometimes for many spacecraft in the planet's magnetosphere. We agree with you that the spacecraft can detect some magnetic field signatures caused by the flapping motion of the magnetotail and highly inclined orbit. One can combine with plasma data to rule out some magnetic structures from the detection if the plasma data is available.

3. The paper does not mention which correlation coefficient is considered. If the Pearson correlation coefficient is calculated, then the study detects the times when there is a linear relation between the data and the TFC such as: data=A*TFC+B (where A and B constants). It is not clear in the paper that this relation is expected. If the purpose is to identify E-R flux ropes features in the data, then why not fitting the specific functions instead of correlating them? It is important to note that maximizing the correlation coefficient is not a fitting method but investigation of the linear relation between the data and the TFC.

AW: We agree with your comments. We calculated the Pearson correlation coefficients in our method, which is different with the fitting on the signal. It is considered that the magnetic structures can be identified as flux ropes when the correlation coefficients

are close to 1. Since we want to identify possible flux ropes from big database and the huge time is required to fit the flux rope mode with the data, we choose to calculate the correlation coefficients that require shorter time.

We revised the introduction of the method in the new version of the manuscript accordingly.

4. How does the path of the spacecraft affect the results? Is the TFC constructed considering only one path of the spacecraft through the flux rope? Then, it is not clear how the results will not be biased due to that. For example, if different path is considered for the same flux rope, what will the method derive and how the correlation coefficients will be affected.

AW: Considering the symmetric of the flux rope model, the different path of the spacecraft will affect the amplitudes and the scale of the TFC. However, this would not affect to identify the flux ropes in our method, but only bring some small errors on the calculation of the scales.

5. As shown in the paper (Figure 3), the TFC is a "part" of the ideal model shown in Figure 2. More specifically, TFC does not include the "edges" where the B field drops to zero. Under which criteria the specific part of the ideal model is selected. How this selection affects the determined scales?

AW: As shown, TFC didn't include the edge when Bz is zero, but include the edge when Bz and By is close to 0. Actually, By is hard to reach zero in the real signal. Thus, we choose one appropriate TFC to identify the flux ropes. The scales are determined by the parameter r in the E-R model. If r is set, the edge of TFC will not affect the determination of the scale in our method.

6. The method correlates the signatures of two components and the total magnitude. Does it matter which two of the three components you use?

AW: We used the bipolar variation component and core field component in the method.
[Figure]

In the magnetotail, bipolar variation component is usually Bz, and core field component is By.

7. As mentioned in the paper, the correlation coefficient could be very large at small scales. Does the study consider the statistical significance of those coefficients? It is also mentioned that the method will be improved in order to be applied to data-sets in the turbulent magnetosheath. Is there a specific plan for the future?

AW: We didn't consider the statistical significance of those coefficients. We considered two ways in order to apply our method in the turbulent plasma: 1) all correlation coefficients of two components and the amplitude of magnetic field should be high (larger than the given threshold) at the same time and the same scale; 2) set the threshold for the amplitude and the scale. If the amplitude is too small, the magnetic structures will be not selected as flux rope.

We plan to statistically survey and investigate the scales and global distribution of flux ropes in the magnetosheath using MMS data.

8. Minor point: It probably worth adding references to previous studies that have used maximizing the correlation coefficient methods.

AW: Thanks for your suggestions. We added some references there.

Please also note the supplement to this comment:
https://www.ann-geophys-discuss.net/angeo-2018-42/angeo-2018-42-AC3-
supplement.pdf
* * *

---

## Author Comment (AC4) · 31 Jul 2018

Dear Dr. AW Smith, thanks for you valuable comments and suggestions that we tried to consider in the re-submission. We have revised and improved the manuscript in response to the comments. All revised parts are marked in red in the text. Detailed answers to the comments are listed below.

Comments: The paper introduces a new technique that could be used to locate magnetic flux ropes within spacecraft magnetometer data. The correlations between signatures in two field components and the total field are used to locate the structures in data. The technique is tested using a model before being applied to example spacecraft data. The method is interesting and potentially very useful, though some of the

ideas could be further developed. Specific Comments: 1) The test with the model and additional random noise (Section 3) could be further used to benchmark the technique. Currently, the level of noise applied is very low and (to the eye) doesn't change the signature significantly. It would be a good test to increase the value of this noise incrementally (e.g. 20%)

AW: Thanks for your suggestions. Considering the high precision and resolution of magnetic field measurement, 10% of the noise is very high. Moreover, even increasing noise would not affect the results because two factor are used there: 1) all correlation coefficients of two components and the amplitude of magnetic field should be high at the same time and the same scale (larger than the given threshold); 2) set the threshold for the amplitude and the scale. If the values are smaller than the threshold, one rules out the possibility.

2) The dependence on the spacecraft trajectory is discussed (Line 232+), but only qualitatively. Simple tests could be performed with magnetic field models to investigate the efficacy of the method with various trajectories. This would significantly help the discussion and justification of the technique.

AW: We discussed effects from the spacecraft trajectory in the real data. Actually, bipolar variation in Bz component heavily depends on the spacecraft trajectory. If there are no bipolar variations in Bz component, the tests may fail because the low correlation coefficients in our opinion (three correlation coefficients of two components and the amplitude of magnetic field should be higher than the given threshold at the same time and the same scale).

3) In general, the work would benefit from additional justification regarding the use of the technique. For example, in what specific ways is the method an improvement over previous attempts/survey methods (e.g. by eye searches)? What is the problem/science question that the use of this technique would help to solve? This discussion is hinted at around Line 252, but could do with development and would improve

the impact of the work.

AW: Thanks for your suggestions. We revised this discussion part in the new version of the manuscript (Line 246-250, 257-261).

Please also note the supplement to this comment:
https://www.ann-geophys-discuss.net/angeo-2018-42/angeo-2018-42-AC4-supplement.pdf

---

## Author Response (AR1)

**Reply to Referee #1 (RC1)**

We greatly thank the reviewer for the valuable comments and suggestions that we tried to consider in the re-submission. We have revised and improved the manuscript in response to the reviewer's comments. All revised parts are marked in red in the text. Detailed answers to the comments are listed below.

**General Comments**

This paper introduces a new method that can be used to search for flux ropes from in-situ observations. This method fits observation data into a flux rope model and uses the correlation coefficient between data and model to tell where the flux rope is. This method is important for making future studies on flux ropes easier. However, there is an important issue that needs to be clarified about this method (see the specific comments below).

**Specific Comments**

Line 182 and all the following parts: When the authors compare the model with real data, they did not mention how they determined the unit length for the model and the B0 to put in the model. These values must be related the data and are crucial for applying the model to the data. Therefore, the authors should explain in detail how these values are determined.

Thanks for the referee's reminding. We used the same unit of the real data as the unit length for the model, i.e. second ('s') in our test. We revised the caption of figures accordingly.

The amplitude  $(B_0)$  in the TFC is determined by the maximum value of  $B_t$  during the interval when calculate correlation coefficients.

**We included this in the Line 184-187.**

Technical corrections

*Line 109: unite/s -> unit/s*

Line 109: "thus set  $a=0.735 \text{ s} \dots$ " Here the authors use the 's' for the unit of a and b, which are length quantities. In other parts of the text the authors use 's' to mean 'seconds', a unit for time. In this line, 's' actually means 'units'. Please use a different letter for this unit.

*Line 191: estimate -> estimated*

Line 196: dynamic -> dynamics

*Line 206: 6b-6d -> 7b-7d*

Figure 4: please explain in the caption that the time scale on the vertical axes is the  $\tau$  mentioned in the text.

According to the referee's suggestions, we have revised the related parts in the new version of the manuscript.

**Reply to Referee #2 (RC2)**

**We greatly thank the reviewer for the valuable comments and suggestions that we tried to consider in the re-submission. All revised parts are marked in red in the text.**

Comments on the manuscript entitled "A new method to identify flux ropes in space plasmas" by Huang et al. Magnetic flux ropes have been regarded as an most important byproduct of magnetic field reconnection. They can raise the reconnection rate, generate energetic electrons, and change the magnetic topology. The typical signatures of a magnetic flux rope are a bipolar variation of magnetic field in the normal direction of the current sheet, and a significant enhancement of the core field as well as the total magnetic field strength. Generally, the flux ropes in the magnetopause and the magnetotail can be easily recognized from the spacecraft measurement by eyes. However, it becomes complicated while the flux ropes were located in the magnetosheath where the current distribution was turbulent. In this paper, the authors proposed a new method to identify the flux ropes in both the large-scale current sheet and the small-scale current sheet. Furthermore, this method was tested by the Cluster and MMS observations in the magnetosheath and magnetotail. The results indicates that the method can select the flux rope effectually. In my view, the result is new and interesting, and suitable to publish in Ann. Geophys. After the following issues are considered. Line 26 flux ropes  $\rightarrow$  magnetic flux ropes Line 31 Wang et al., PRL 2010 reported the first evidence of magnetic flux rope or island generated during a single X-line, should be cited here. Line 34, Wang et al., Nat. Phys. 2016 shows a clear picture of the dissipation role that the magnetic flux ropes play on. Line 103 cross  $\rightarrow$  crosses

**We have revised manuscript according to the referee's suggestions.**

**1 A new method to identify flux ropes in space plasmas**

Shiyong Huang1, Pufan Zhao1, Jiansen He2, Zhigang Yuan1, Meng Zhou3, Huishan
Fu4, Xiaohua Deng3, Ye Pang3, Dedong Wang1, Xiongdong Yu1, Haimeng Li4, Roy
Torbert5, and James Burch6

[revised manuscript text omitted]

4. The correlation coefficients at small scale (especially in  $B_y$  and  $B_t$ ) could be very large, which may affect our results. The method may find some possible structures related to such fluctuations. We will improve this method and apply it to detect the flux ropes in the turbulent magnetosheath in the future.

**284 Acknowledgement**

We thank the entire Cluster and MMS team and instrument leads for data access and 286 support. This work was supported by the National Natural Science Foundation of 287 China (41404132, 41574168, 41674161), and China Postdoctoral Science Foundation 288 Funded Project (2015T80830). SYH acknowledges the support by Young Elite 289 Scientists Sponsorship Program by CAST (2017QNRC001). MMS Data is publicly 290 available from the MMS Science Data Center at http://lasp.colorado.edu/mms/sdc/. Cluster Data is publicly available from the Cluster Science Archive at 291 292 http://www.cosmos.esa.int/web/csa.

**294 **Reference**

[revised manuscript text omitted]